# Functional and Bioactive Benefits of Selected Microalgal Hydrolysates Assessed In Silico and In Vitro

**DOI:** 10.3390/md23020053

**Published:** 2025-01-22

**Authors:** Elena Aurino, Leticia Mora, Antonio Marzocchella, Christina M. Kuchendorf, Bärbel Ackermann, Maria Hayes

**Affiliations:** 1Dipartimento di Ingegneria Chimica, dei Materiali, e della Produzione Industriale, Università degli Studi di Napoli ‘Federico II’, Piazzale Tecchio, 80, 80125 Naples, Italy; elena.aurino@unina.it (E.A.); marzocch@unina.it (A.M.); 2Food BioSciences, Teagasc Food Research Centre, Ashtown, D15 DY05 Dublin, Ireland; 3Instituto de Agroquímica y Tecnología de Alimentos (CSIC), Avenue Agustín Escardino 7, 46980 Paterna, Spain; lemoso@iata.csic.es; 4Institute of Bio- and Geosciences, IBG-2—Plant Sciences Forschungszentrum Jülich GmbH, Wilhelm-Johnen-Str., 52428 Jülich, Germany; c.kuchendorf@fz-juelich.de; 5Stadt Erftstadt, Stabsstelle Klimaschutz, Holzdamm 10, 50374 Erftstadt, Germany; baerbel.ackermann@erftstadt.de

**Keywords:** microalgae, hydrolysis, bioactive peptides, anti-inflammatory, anti-hypertensive, anti-oxidative, health, techno-functional activities, foaming, oil holding capacity

## Abstract

BIOPEP-UWM, a peptide database, contains 5128 peptides from a myriad of resources. Five listed peptides are Angiotensin-I-converting enzyme (ACE-1; EC3.4.15.1) inhibitory peptides derived from a red alga, while two from *Chlorella vulgaris* have anti-cancer and antioxidative bioactivities. Herein, we describe a process combining hydrolysis with two enzymes, Alcalase and Viscozyme, and filtration to generate protein-rich, bioactive peptide-containing hydrolysates from mixed species of *Chlorella* sp. and *Scenedesmus* sp. The potential of generated algal hydrolysates to act as food ingredients was determined by assessment of their techno-functional (foaming, emulsification, solubility, water holding, and oil holding capacity) properties. Bioactive screening of hydrolysates in vitro combined with mass spectrometry (MS) and in silico predictions identified bioactive and functional hydrolysates and six novel peptides. Peptides derived from *Chlorella* mix have the sequences YDYIGNNPAKGGLF and YIGNNPAKGGLF with predicted anti-inflammatory (medium confidence) and umami potential. Peptides from *Scenedesmus* mix have sequences IEWYGPDRPKFL, RSPTGEIIFGGETM, TVQIPGGERVPFLF, and IEWYGPDRPKFLGPF with predicted anti-inflammatory, anti-diabetic, and umami attributes. Such microalgal hydrolysates could provide essential amino acids to consumers as well as tertiary health benefits to improve human global health.

## 1. Introduction

Proteins are necessary for adequate human nutrition. Global demand for protein is expected to double by 2050, mainly due to population growth and changes in dietary habits [1]. Most proteins traditionally consumed are of animal origin, such as meat, eggs, and dairy products [2]. The production of traditional protein sources requires large areas of land for animal livestock and feed production, which contributes to deforestation in favor of agricultural land. Moreover, the production of animal feed requires high water and nitrogen consumption for fertilizers [2]. In addition, emissions from livestock farming also have a negative impact on the environment, and the production of meat proteins is known to contribute to greenhouse gas emissions (GHGs). Currently, approximately 12% of GHG emissions result from livestock farming [3]. Satisfying the world’s growing demand for proteins only using animal products does not appear to be sustainable, which is why researchers have set out to find new, alternative proteins.

Microalgae are a source of protein with potential for use in foods to provide adequate nutrition. They are known to be rich in proteins [4,5] containing all the essential amino acids needed for human growth and nutrition [5]. In addition to their nutritional value, microalgae can synthesize bioactive molecules such as peptides, vitamins, and antioxidant compounds [5]. Species such as *Chlorella vulgaris*, *Scenedesmus obliquus,* and *Spirulina platensis* may contain between 51 and 58%, 48 and 60%, and 71% proteins, respectively, on a dry weight basis [6]. Microalgal protein productivity can reach 4–15 tonnes of protein/ha/year compared to other terrestrial crops, including soybean, pulse legumes, and wheat, which can yield 0.6–1.2 tons/ha/year, 1–2 tonnes/ha/year, and 1.1 tonnes/ha/year, respectively [7]. Moreover, microalgae sequester carbon dioxide, converting it to organic carbon through photosynthesis [8].

However, microalgal digestibility values are generally low in comparison to proteins of animal origin like dairy, fish, and beef and plant proteins like soy [9]. The low digestibility of microalgal proteins affects their uptake in the human or animal body. It is mainly caused by the tough cell walls of microalgae, most of which are indigestible to monogastric animals and humans [10]. The best way to improve cell digestibility is to apply cell disruption techniques to the biomass to make the proteins more readily available. Therefore, the digestibility value of algal proteins depends on the downstream process applied to the biomass [11]. To confirm this statement, several studies highlight an increase in the digestibility of treated microalgal biomass compared to whole biomass. For example, Wang and colleagues reported an increase in the digestibility of the disrupted biomass of *Chlorella vulgaris* and *Chlorella sorokinina* after cell wall disruption using micro-fluidization with respect to the untreated biomasses [12]. A similar result was reported for *Nannochloropsis oceanica, C. vulgaris, and Tetraselmis* sp. after mechanical and enzymatic processes [13]. The same results were obtained for recalcitrant cells such as *Galdieria sulphuraria* treated with Viscozyme [13]. Other than digestibility, some properties of microalgae are detrimental to their incorporation in food products. Nowadays, the use of microalgae biomass as ingredients is hampered by their powdery texture, dark green color, and slightly fishy smell [14]. The addition of more than 10% whole microalgae into a food product negatively affects the sensory properties, changing the taste and the texture of the final product [2]. Again, applying a refining step to microalgae may help to mitigate these deleterious properties of whole, dry biomass and improve the organoleptic features of microalgae ingredients. It is important to note that protein extraction and food processing are often accompanied by protein denaturation, which affects protein bioactivity [15]. On this basis, it is mandatory to study, in depth, the techno-functional properties of algal extracts.

One of the mildest methods to treat microalgae biomass in order to release protein is to produce a hydrolysate by using enzymes [16]. Other than mildness and ease of scalability, the points of strength of this technique are low energy consumption and low capital investments. On the other hand, it requires a long production time, and the enzyme costs can be high [17]. The employment of enzymes such as lipase, phospholipase, protease, and cellulases on *C. vulgaris* resulted in a protein extract containing a protein content of 68% [18]. To offset processing costs, it is important to expand the market value of hydrolysates. This is possible by identifying the tertiary benefits (health and functional) of the peptides found in end-process hydrolysates. Microalgal hydrolysates rich in protein can be used actively in food formulations due to their techno-functional properties. As sources rich in protein, they are good candidates to act as meat substitutes due to their ability to enhance mouth feel, emulsion and foam stability and shelf life and they can find applications as ingredients for products including ice cream, mayonnaise, and creams [19]. Techno-functional properties of microalgal hydrolysates can influence organoleptic characteristics such as texture and mouthfeel, and they can impact physicochemical properties such as foaming, emulsifying, solubility, and water and oil holding capacity [20]. Microalgal proteins are mostly employed for their promising functionality as foaming and emulsifying agents at neutral pH [21].

Bioactive peptides, often generated during the manufacture of protein hydrolysates, have numerous health-promoting benefits, including antioxidant, antihypertensive, and anti-cancer properties [22]. Peptides are formed from proteins after enzymatic hydrolysis, which can occur through gastrointestinal digestion, fermentation, or the use of exogenous enzymes in food processing [23]. Each proteolytic enzyme has a preferential cleavage site for the selected protein, making the enzyme choice relevant [24]. The antioxidant activities of microalgal protein extracts were previously reported. For example, the free radical scavenging activities of peptides extracted from *Chlorella ellipsoidea* towards DPPH, hydroxyl, and peroxyl radicals were investigated and compared with the activity of synthetic antioxidants such as BHT [25,26]. Both studies found that the antioxidant activity of microalgal-derived peptides and the synthetic antioxidant were similar, suggesting that *C. ellipsoidea* peptides are excellent antioxidants.

Angiotensin-converting enzyme (ACE-1) inhibitory activities were also reported previously for microalgal-derived peptides. ACE-1 is responsible for vasoconstriction, which, together with oxidative stress, is the main cause of hypertension and cardiovascular disease [27]. Peptides extracted from *C. vulgaris* [28,29] displayed ACE-1 inhibitory activities previously. In particular, the ACE-1 inhibition capacity of different strains of *Chlorella* species was identified previously in several studies [30,31,32,33]. In addition, the anti-diabetic (T2D inhibition) potential of several microalgae and peptides was also investigated recently, and results indicate that they may be candidates for product development for prevention of T2D.

The aim of this work was to produce bioactive peptide-containing protein hydrolysates from the selected microalgae *Chlorella* sp. and *Scenedesmus* sp. using a two-step hydrolysis process. The potential health benefits and techno-functional attributes of generated hydrolysates were assessed in vitro using a selection of bioassays and techno-functional assays. The bioactive peptides contained in bioactive hydrolysates were characterized using mass spectrometry, and in silico prediction analysis as described previously [34,35,36,37,38] was used to determine additional health benefits for selected peptides.

## 2. Results

Several studies performed enzymatic hydrolysis on *Chlorella* sp. and *Scenedesmus* sp. algal strains previously [39,40]. Enzymatic hydrolysis is a mild extraction method used mostly to extract proteins and peptides and to preserve extract features and health benefits as much as possible [41]. The enzymes employed in our work included a carbohydrase, capable of digesting the carbonaceous cell wall of algal and vegetable cells, and a protease, able to degrade the proteins to peptides. To add value to a high-cost protein extraction process, it is important to offset processing costs by expanding potential market applications of the end product. Our results show that the developed microalgae/mix hydrolysates contain novel peptides with predicted anti-inflammatory activities identified using in vitro Cyclooxygenase inhibition assays. 

### 2.1. Hydrolysate Generation and Proximate Analysis

A two-step enzymatic hydrolysis process using the commercially available enzymes Viscozyme and Alcalase was applied to the dried, microalgal biomass, which was supplied by the IDEA project partner FZJ Jülich, Germany. Viscozyme, a carbohydrase, was used for initial hydrolysis to partially break the algal cell wall and release the intracellular material, including proteins. Alcalase was then applied to break down proteins into bioactive peptides. After hydrolysis with Viscozyme, a degree of hydrolysis (DH) of 25.4% and 27.46% was obtained following application to *Chlorella* mix and *Scenedesmus* mix, respectively. Subsequent hydrolysis with the Alcalase enzyme resulted in a DH of 48.44% and 46.35% when applied to *Chlorella* mix and *Scenedesmus* mix, respectively. The results obtained in this work for the *Scenedesmus* mix biomass are coherent with results obtained by Romero-García and their co-worker [42] that performed a 3-step enzymatic hydrolysis procedure on *Scenedesmus almeriensis* using Viscozyme, Alcalase and Flavourzyme and obtained a DH of 42% (Table 1).

The proximate compositional analysis results for whole algal biomass, the whole hydrolysates, and the permeate and retentate fractions generated in the two-step hydrolysis process are shown in Table 2.

The protein content of both 3 kDa molecular weight cut-off (MWCO) permeates generated independently from the two algal strains was significantly greater than the protein content of the starting biomass. The permeate protein content was 44.73% and 41.14% for *Chlorella* mix and *Scenedesmus* mix, respectively, larger than the starting protein content for whole biomasses of 38.02% and 37.81% for *Chlorella* mix and *Scenedesmus* mix, respectively. Filtration with 3 kDa MWCO membranes also reduced the fat content of both algal 3 kDa-permeates compared to the whole biomasses and hydrolysates, respectively. The ash content of both 3 kDa permeates was significantly greater than the whole biomass or generated hydrolysates. For example, the ash content of the *Chlorella* mix 3 kDa permeate (10.10% ash) fraction almost doubled compared to the whole biomass (4.85% ash) and generated hydrolysate (5.27% ash), respectively. An increase in ash content was also observed for the 3 kDa *Scenedesmus* mix permeate, where the ash content was determined to be 7.50% compared to the whole biomass that was found to contain 2.03% ash and the *Scenedesmus* mix hydrolysate, which was found to contain 4.53% ash.

### 2.2. Techno-Functional Analysis

#### 2.2.1. Protein Solubility

Not all the proteins extracted from microalgal biomass have a positive effect on the techno-functional properties of the ingredient. Only the soluble fraction of protein works at the oil–water interface. Therefore, to see the efficiency of the extraction, it is necessary to assess the percentage of soluble proteins in the extract. The pH of the solution affects the solubility of the proteins. At their isoelectric point (pI), proteins show a neutral net charge, making them less soluble in water [43]. The protein solubility of generated microalgal hydrolysates was performed at different pH values; the results obtained are reported in Figure 1. The statistical analysis on the data set reveals a non-significant difference between values obtained at different pH levels; therefore, a significant minimum in solubility value can be determined for either of the two investigated hydrolysates. The only almost significant difference was found between the two biomasses considered at pH 2 (*p* = 0.068).

#### 2.2.2. Water Holding Capacity (WHC) and Oil Holding Capacity (OHC)

Water holding capacity and oil holding capacity (WHC and OHC) refer to the ability of an ingredient to retain water or oil, respectively. A high capability to retain water is an indicator of a better shelf life for a final product/ingredient because it relates to limited moisture loss during storage [44]. On the customer acceptability side, good WHC improves the texture of food products and ensures thickness and viscosity of the final product [45]. A high OHC improves the palatability of an ingredient/product, and it is useful for retention of flavors in the food product [46]. A high WHC reflects a lower concentration of water-soluble proteins [47]. The WHC results determined for *Chlorella* mix and *Scenedesmus* mix in this study were 1.83 and 1.78 g of water/g of hydrolysate, respectively (Figure 2). The WHC of the microalgal hydrolysates examined in this work is less than those reported previously for other whole algae. For example, *Spirulina platensis* demonstrated a WHC of between 2.53 and 16.6 g of water/g of sample. Variations observed related to the pH of the sample [48]. In addition, the WHC of the *Porphyridium cruentum* species was previously reported as 8.16 g of water/g of sample [49]. There is no statistical difference between the OHC of the two studied biomass hydrolysates, and data do not show a significant difference between the OHC when different oils were used (Figure 2).

The OHC is an index of the non-polar protein fractions’ capacity to bind oil [50]. In this work, sunflower oil and rapeseed oil were used to determine the OHC of the microalgal hydrolysates. *Chlorella* mix hydrolysate had an OHC of 5.12 g of sunflower oil/g of hydrolysate and 4.24 g of rapeseed oil/g of hydrolysate. *Scenedesmus* mix hydrolysate had an OHC of 5.32 g of sunflower oil/g of hydrolysate and 4.88 g of rapeseed oil/g of hydrolysate. *Scenedesmus* mix hydrolysate demonstrated greater OHC than the *Chlorella* mix hydrolysate, especially for rapeseed oil (Figure 2), but this was not statistically significant. Oil-holding capacity (OHC) refers to the ability of a protein or hydrolysate to bind with lipids and influences sensory characteristics of foods, in particular, the mouthfeel and flavor of foods. The OHC values identified for *Scenedesmus* sp. mix and *Chlorella* sp. mix are comparable to chickpea and cottonseed protein isolates but are less than values reported for yellow pea and chickpea protein concentrates made using isoelectric precipitation [50,51].

#### 2.2.3. Emulsion Activity and Heat Stability

The emulsion activity (EA) and the emulsion heat stability (EHS) values of both microalgal hydrolysates were evaluated. EA and EHS were evaluated at different pH levels between 2 and 10, as emulsion properties are influenced by solubility, which in turn is influenced by pH. The obtained results are reported in Figure 3.

### 2.3. Bioactivity Assessment of 3 kDa Microalgal Permeates

#### 2.3.1. ACE-1 Inhibition

The bioactivities of generated microalgal 3 kDa permeates were assessed in vitro using a suite of bioassays, including the alpha-amylase inhibition assay, the Angiotensin-1-converting enzyme (ACE-1) inhibition assay, and additionally, permeates were assessed for their potential antioxidant activities using the ABTS scavenging assay. The studied 3 kDa algal permeates demonstrated ACE-1 inhibitory activity in vitro. When assayed at a concentration of 1 mg/mL, the permeates from *Chlorella* mix hydrolysate and *Scenedesmus* mix hydrolysates inhibited ACE-1 by 88.07 ± 1.69% and 86.24 ± 2.89%, respectively, compared to the commercial control Captopril^©^ assayed at a concentration of 0.05 mM, which inhibited ACE-1 by 97.88%. Previous studies by Li and colleagues [51] demonstrated the ACE-1 inhibitory effect of a 3 kDa *Chlorella pyrenoidosa* permeate where the activity reported was 84.2 ± 0.37% [51]. In addition, Tejano and colleagues previously reported that a 5 kDa protein fraction from *C. sorokiniana* inhibited ACE-1 by 34.29 ± 3.45% [52]. Both microalgal hydrolysates generated in this work displayed greater inhibition of ACE-1 than results reported previously.

#### 2.3.2. ABTS Radical Scavenging Effect

The antioxidant effect of generated microalgal-3 kDa permeates was assessed using the ABST scavenging assay. When both algal permeates were assessed at 1 mg/mL concentrations using the positive control resveratrol, the ABTS-antioxidant activity percentage values obtained were 72.54% and 76.17% for the 3 kDa permeates generated from the *Chlorella* mix and *Scenedesmus* mix hydrolysates, respectively (Table 3). The antioxidant activities of microalgal hydrolysates were reported previously. Alzahrani and colleagues assessed the antioxidant activity of *Chlorella vulgaris* biomass hydrolyzed with Alcalase previously. This study identified an ABTS radical scavenging activity value of 63% [53]. Additionally, Afify and colleagues report an ABTS radical scavenging activity for *S. obliquus* hydrolyzed with Papain of 87.03% [54].

#### 2.3.3. α-Amylase Inhibitory Activity

The α-amylase inhibitory potential of generated algal 3 kDa permeates was also assessed, and a 71.32 ± 12.30% value was obtained for the *Chlorella* mix 3 kDa permeate fraction. Table 3 collates the bioactivity results obtained for the 3 kDa permeate fractions generated from *Chlorella* mix and *Scenedesmus* mix Viscozyme and Alcalase hydrolysates, respectively.

### 2.4. MS and In Silico Analysis of 3 kDa Permeate Fractions Generated from Chlorella Mix and Scenedesmus Mix Viscozyme and Alcalase Hydrolysates

MS sequencing results of the 3 kDa permeate generated from *Chlorella* mix Viscozyme and Alcalase hydrolysate identified thirty peptides with confidence levels ranging from 16.67 to 99.9%. Only two peptides had confidence levels of 95–99.9%. These peptides had the sequences YDYIGNNPAKGGLF and YIGNNPAKGGLF. 284 peptides were identified from the 3 kDa permeate fraction generated from the *Scenedesmus* mix Viscozyme and Alcalase hydrolysate, and only four peptides had confidence levels equal to 99.9%. Peptides identified from the *Scenedesmus* mix 3 kDa permeate included those with the sequences IEWYGPDRPKFL, RSPTGEIIFGETM, TVQIPGGERVPFLF, and IEWYGPDRPKFLGPF. Further details regarding the protein origin of these peptides are shown in Table 4.

In silico analysis was carried out on identified peptide sequences to predict further bioactivities associated with these peptides. Peptides were ranked in terms of their potential bioactivities using an in silico approach published by our group previously [55]. Briefly, the peptide ranker program was used to predict the potential bioactivity of peptide sequences based on amino acid charge and peptide structure [56]. PreAIP http://kurata14.bio.kyutech.ac.jp/PreAIP/ (accessed on 10 August 2024) [57] was used to predict potential anti-inflammatory activity of individual peptides and the collected group of peptides. In Table 4, scores ≥ 0.468 indicate high confidence that the selected algal-derived peptide is anti-inflammatory, values between 0.468 and ≥ 0.388 indicate medium confidence that the peptide is anti-inflammatory, and values of 0.388 to 0.342 indicate that the peptide has a low probability of having anti-inflammatory activity [57]. The novelty of the algal-derived peptides was determined by a search in the database BIOPEP-UWM, and the potential of peptides to impart umami flavors was predicted using Umami-MRNN https://umami-mrnn.herokuapp.com/ (accessed on the 10 August 2024) [58]. In addition, the umami and bitterness of the peptides identified from algae and listed in Table 4 were assessed using Umami_YYDS 2.0 BETA, an umami/bitterness judgment model based on machine learning and chemical descriptors. The peptide YDYIGNNPAKGGLF classifier is umami and the probability is 1.0, the peptide IEWYGPDRPKFL classifier is umami and the probability is also 1.0, and the peptide TVQIPGGERVPFLF classifier is umami, with the probability being 0.8326 (accessed on 30 December 2024).

AntiDMPpred, http://i.uestc.edu.cn/AntiDMPpred/cgi-bin/AntiDMPpred.pl (accessed on 10 August 2024), was used to identify the anti-diabetes type 2 inhibitory potential of microalgal-derived peptides [59].

## 3. Discussion

### 3.1. Hydrolysate Generation

The commercial enzymes Alcalase and Viscozyme were combined and used to generate protein hydrolysates from the microalgal mixed cultures supplied to Teagasc by FZJ-Jülich. This enzyme combination was selected based on previous work carried out by Naseri and colleagues [60], who extracted over 90% of the protein from the red seaweed *Palmaria palmata* using this combination of enzymes followed by alkaline extraction.

Our work increased the content of protein by employing MWCO filtration post hydrolysis. The protein content increased from 38.02% to 44.73% for *Chlorella* mix and from 37.81% to 41.14% for *Scenedesmus* mix, respectively, when MWCO filtration was applied to the individual hydrolysates. This result suggests that many of the proteins and peptides generated during hydrolysis were greater than 3 kDa in size (80.22% for *Chlorella* mix retentate and 71.77% for *Scenedesmus* mix retentate, respectively). Additionally, most of the fat fraction of algal hydrolysates remained in the retentate fraction.

### 3.2. Techno-Functional Activities

#### 3.2.1. Solubility

Both hydrolysates had solubility values similar to those reported previously for algal extracts. Both hydrolysates had solubility values similar to those reported previously for algal extracts. Since the *Chlorella* mix protein solubility at pH 2 is higher than the *Scenedesmus* mix, it is likely that the former protein extract is more like the protein casein in terms of solubility than the latter. Casein, at low pH values like pH 2, has a net positive charge due to protonation of basic side chains (—NH3 +) [61]. A clear difference was not observed at different pH values between the *Scenedesmus* mix and *Chlorella* mix. Microalgae proteins examined in other studies show low solubility at pH < 5, and solubility increases from neutral to alkaline pH (50), as is the case in our work. Protein solubility controls gelation, foaming, and emulsifying capacity and is a measure of food quality in relation to sedimentation, viscosity, and turbidity. Results in Figure 1 suggest that the hydrolysates have a broad solubility range and could be a good fit for food formulations.

#### 3.2.2. Water and Oil Holding Capacities of Microalgal Hydrolysates

Oil and water holding capacities impact the flavor, texture, and mouthfeel of food products. A high WHC value can delay staling of food products and can help to preserve moisture and freshness in baked goods [62]. A high OHC can be useful in the production of products like mayonnaise. WHCs for *Chlorella* protein extracts reported previously ranged from 1.7 to 5.1 g of water/g of protein hydrolysate. The values obtained for the *Chlorella* mix and *Scenedesmus* mix hydrolysates in this study compare favorably (1.83 g of water/g of protein hydrolysate and 1.78 g of water/g of protein hydrolysate) [62].

OHC values observed ranged from 5.12 g of sunflower oil/g of hydrolysate to 5.32 g of sunflower oil/g of hydrolysate. These values compare favorably with OHC values for *Spirulina platensis* protein hydrolysates, which range from 5.3 to 7 g of soybean oil per gram of protein hydrolysates [63].

#### 3.2.3. Emulsion Activity and Stability

Amphiphilic proteins can be used as emulsifiers in food systems to reduce interfacial tension and stabilize the oil–water interface. Oil-in-water emulsions are found in milk, salad dressings, ice cream, and butter, and may be used as delivery systems to protect functional ingredients like omega-3 oils [64]. Both protein hydrolysates demonstrated the highest emulsification activities at alkaline pH values in sunflower oil (pH 8 for *Chlorella* mix and pH 10 for *Scenedesmus* mix).

Both hydrolysates demonstrate optimum emulsion activities at pH 10 and emulsion stabilities at pH 10 and 6 for *Chlorella* mix hydrolysate and *Scenedesmus* mix hydrolysate, respectively. In other works, the best emulsifying capacity of *Chlorella vulgaris* proteins was obtained at pH 7 (50). The pH value for which emulsion activity is optimal determines the use of algae in food products (50). Emulsions that are stable at neutral or alkaline pH values, as is the case here, are suited to neutral/basic food preparations, usually preparations like alkaline drinks containing fruit or vegetables or nut “dairy alternative” drinks. They could also find use as emulsifiers, for example, in salad dressings or mayonnaise.

### 3.3. Bioactivity Assessments

#### 3.3.1. ACE-1 Inhibition

The observed ACE-1 inhibitory benefits of the 3 kDa fractions are reported in Table 3. When assayed at concentrations of 1 mg/mL, *Chlorella* mix 3 kDa permeate and *Scenedesmus* mix 3 kDa permeate inhibited ACE-1 by 88.07% and 86.04%, respectively. MS work combined with in silico work identified a number of novel bioactive peptides in these permeate fractions (Table 4). Many of the peptides listed have bulky-side-chain aromatic amino acids at the C-terminal of the peptide sequence, and this is an indicator of ACE-1 inhibitory activity [65]. The ACE-1 inhibitory values obtained for both hydrolysates combined with the identified peptides in Table 4 highlight the potential of these hydrolysates to have anti-hypertensive activities as they compare favorably to Captopril©, the antihypertensive drug, in terms of ACE-1 inhibition. Further in vivo trials in Spontaneously Hypertensive Rats (SHRs) would confirm an antihypertensive effect.

#### 3.3.2. Alpha-Amylase Inhibition

The α-amylase inhibitory potential of generated algal 3 kDa permeates was determined as 71.32 ± 12.30% for the *Chlorella* mix 3 kDa permeate fraction. This result is not surprising, as algae have garnered great interest as a source of new drugs for the prevention of type-2 diabetes (T2D). Indeed, a recent study by Sahoo and colleagues found active compounds in methanolic extracts from *Chlorella vulgaris*. These extracts had α-amylase IC_50_ values of 2.66 µg/mL compared to a positive control, Acarbose, that had an IC_50_ value of 2.85 µg/mL [66]. Acidified methanol extractions perform best for proteomics analysis [67], so it is likely that peptides are inhibiting α-amylase in the methanol extracts generated by Sahoo and colleagues, and these could be similar to those identified herein [66]. Use of these microalgal hydrolysates is a potential strategy and therapeutic approach to reduce blood glucose levels in the management of T2D.

#### 3.3.3. Antioxidant Activities

ABTS-antioxidant activity percentage values of 72.54% and 76.17% were identified for *Chlorella* mix and *Scenedesmus* mix hydrolysates, respectively. Previously, the antioxidant peptide with amino acid sequence VECYGPNRPQF was identified from *Chlorella vulgaris* [68]. The antioxidant activity of identified peptides from both hydrolysates was not explored further as part of this paper, and the peptide VECYGPNRPQF was not identified in the *Chlorella* mix hydrolysate. A recent study concerning *Scenedesmus obliquus* protein hydrolysates identified their potential for use as antioxidative, functional food ingredients [69]; however, the identified peptides in this work were not characterized for antioxidant potential in this work.

### 3.4. MS and In Silico Analysis

Over 70 peptides were identified from both algal 3 kDa permeate fractions; however, few had protein sequence similarity percentage values greater than 95%, and those that did, are listed in Table 3. The identified peptides have sequence similarity to photosystem proteins (peptides YDYIGNNPAKGGLF and YIGNNPAKGGLF derived from the *Chlorella* mix hydrolysate and peptide RSPTGEIIFGGETM derived from the *Scenedesmus* mix hydrolysate). The photosystem protein-derived peptides were identified as potential anti-inflammatory and anti-diabetic peptides, particularly the peptide RSPTGEIIFGGETM derived from the *Scenedesmus* mix hydrolysate. This peptide was identified as having medium potential anti-inflammatory activity and potential to be a T2D enzyme inhibitor using the programs PreAIP http://kurata14.bio.kyutech.ac.jp/PreAIP/ (accessed on 10 August 2024) and http://i.uestc.edu.cn/AntiDMPpred/cgi-bin/AntiDMPpred.pl (accessed on 10 August 2024), respectively. Peptides identified by MS are novel but are derived from known photosystem proteins from microalgae. In silico analysis of the identified peptides highlights further the added value of protein hydrolysates in terms of the creation of tertiary health benefits that go above and beyond basic human nutrition. From in silico analysis, the peptide TVQIPGGERVPFLF is likely to have anti-diabetic activity, further validating the observed α-amylase inhibitory activity identified in vitro for the hydrolysate. Moreover, several of the peptides are predicted to impart an umami taste, including RSPTGEIIFGGETM from the *Scenedesmus* mix hydrolysate. Umami peptides with antioxidant properties may be used as natural preservatives in meat products, sauces, soups, and snacks to extend the shelf life of foods.

### 3.5. Cyclooxygenase-2 (COX-2) Inhibitory Activity

Two peptides, peptide IEWYGPDRPKFSPF and TVQIPGGERVPFLF, were chemically synthesized and tested for their anti-inflammatory activity using the Cyclooxygenase-2 (COX-2; EC 1.14.99.1) inhibition assay. Peptides were assessed for their ability to inhibit COX-2 at a concentration of 1 mg/mL compared to a positive control (Diclofenc—a commercial drug). Both peptides inhibited COX-2 minimally—by 23.07% and 20.39%, respectively. It is likely that these peptides have a different mechanism of anti-inflammatory/anti-pain action, and they warrant further examination using assays including Monoacylglycerol lipase (MAGL) inhibition as well as cell line work examining the up- or down-regulation of inflammatory markers like IL-6, TNF-α, and others.

## 4. Materials and Methods

### 4.1. Biomass

FZJ Jülich, Germany, as part of the IDEA project, supplied two mixed algal biomass samples to Teagasc. The first sample, harvested in June 2018, called *Chlorella* mix, was composed of *Chlorella* sp. (92.5% *w*/*w*), *Scenedesmus* sp. (5% *w*/*w*), and *Chlorocococcus* sp. (2.5% *w*/*w*). The second sample, harvested in May 2018, termed *Scenedesmus* mix, is mainly composed of *Scenedesmus* sp. and included *Chlorella* sp. and other diatoms (Table 5). All biomass was provided in a spray-dried powder form. The composition of each sample was determined by microscopy by Bärbel Ackermann (FZJ, Julich, Germany).

### 4.2. Chemicals

The following enzymes and chemicals were used to generate hydrolysates and characterize hydrolysates and permeate generated from microalgae. The enzymes Viscozyme and Alcalase were supplied by Merck (Dublin, Ireland), as well as trichloroacetic acid (TCA), dimethylsulfoxide (DMSO), resveratrol, and Captopril©. NBS Biologicals Ltd. (Cambridgeshire, England, UK) supplied the ACE-1 inhibition assay kit. Zen-Bio, Inc. (Research Triangle Park, NC, USA) supplied the ABTS antioxidant kit, and the α-amylase kit was purchased from Abcam (Cambridge, UK).

### 4.3. Protein Extraction Using Hydrolysis

To obtain algal hydrolysates (Figure 4), freeze-dried microalgal mixtures were suspended in ddH_2_O (10% *w*/*w*) and placed in a Grant JB Aqua 12 water bath (Grant instrument, Royston, England, UK) at 80 °C for 10 min to deactivate the enzymes already present in the biomass. After adjusting the pH using 1M HCl to obtain a pH between 3 and 5, Viscozyme enzyme (1% *w*/*w*) was added to the sample, and the mixture was subsequently incubated at 45 °C × 2 h × 220 rpm. The samples were heat-deactivated in a water bath at 80 °C × 10 min. The same procedure was performed subsequently with the enzyme Alcalase post adjustment of the pH using 1 M NaOH to obtain a pH of 8–8.5. Hydrolysates were filtered through a 3 kDa cellulose cartridge filter (Merck Millipore, Cork, Ireland) with a molecular weight cut-off (MWCO) of 3 kDa to generate a permeate fraction. The 3 kDa Millipore Prep/Scale Tangible Flow Filtration (TFF) system (Merck Millipore, Cork, Ireland) was employed for this purpose. Deionized water (Nano-water purification and filtration system, Merck, Dublin, Ireland) was used to flush the permeate from the system at a flow rate of 10 mL/minute.

To assess the efficiency of the enzymatic process, the permeate yield, the protein recovery, and the degree of hydrolysis were calculated as follows:$$\%\;permeate\;yield\;\left[\frac{g}{g}\right]\;=\;\frac{mass\;of\;permeate}{mass\;of\;whole\;biomass}\;\times \;100$$
and$$\%\;protein\;recovery\;[g/g]\;=\;\frac{mass\;of\;protein\;in\;the\;permeate}{mass\;of\;protein\;in\;the\;whole\;biomass}\;\times \;100$$

The degree of hydrolysis (DH) was calculated at the end of each hydrolysis stage using the TCA method described by Hoyle [70]. A sample of 1 mL of hydrolysate was collected after the deactivation step and added to 1 mL of 20% (*w*/*v*) trichloroacetic acid (TCA). The solutions were left to settle for 30 min and then centrifuged at 7800× *g* 15 min. The percentage of proteins in the supernatant and the hydrolysate sample was assessed using the QuantiPro BCA Assay kit (Sigma, St. Louis, MO, USA) as per the manufacturer’s instructions, so as to calculate the DH:$$DH\%\;=\;\frac{TCAsoluble\;N\;in\;the\;supernatant}{total\;N\;in\;the\;hydrolysate}\;\times \;100$$

The hydrolysate obtained was filtrated using a 3 kDa molecular weight cut-off (MWCO) membrane filter (Millipore, Tullagreen, Carrigtwohill, Co. Cork, Ireland), obtaining a permeate and a retentate. The extraction was performed in duplicate for each microalgae strain.

### 4.4. Proximate Compositional Analysis

Proximate analysis was performed on fractions generated at each step of the protein extraction process (Figure 4). Whole microalgae, whole hydrolysates, the retentate, and permeate samples were frozen at −80 °C and freeze-dried using an industrial-scale FD 80 freeze-drier (Cuddon Engineering, Marlborough, New Zealand) prior to analyses. The protein content was obtained based on the nitrogen content of the samples. The nitrogen percentage in the samples was determined using the LECO FP628 Protein analyzer (LECO Corp., St. Joseph, MI, USA) with the Dumas method, according to AOAC method 992.15 (1990) [71]. The protein content was calculated using a conversion factor of 6.25. To determine the moisture content of the samples, the difference in weights after drying the samples in a Gallenkamp Air Oven set at 105 °C overnight was measured. Fat content was assessed using the Oracle NMR Smart Trac rapid fat analyzer (CEM Corporation, Matthews, NC, USA) using AOAC official methods 985.14 as described previously. The ash content of samples was determined by measuring the difference in weights of samples post drying in a furnace at 600 °C overnight.

### 4.5. Mass Spectrometry in Tandem Analysis

A total of 1 µL of every sample was quantified by nanoDrop (A 280 nm, e = 1 mg/mL). For every sample, the volume corresponding to 190 ng was diluted in 20 µL with 0.1% FA and loaded in an Evotip pure tip (EvoSep) according to the manufacturer’s instructions. Tandem mass spectrometry analysis (LC–MS/MS) was performed in a Tims TOF fleX mass spectrometer (Bruker, Kontich, Belgium). The sample loaded in the Evotip pure tip (Odense, Denmark) was eluted to an analytical column (EvoSep 15 cm × 150 µm, 1.5 µm; Evosep) by the Evosep One system and solved with the 30 SPD chromatographic method defined by the manufacturer. Eluted peptides were ionized in a captive spray with 1700 V at 200 °C and analyzed using the ddaPASEF mode with the following settings: TIMS settings: mode: custom; 1/K0: 0.7–1.76 V·s/cm^2^; ramp time: 100 ms; duty cycle: 100%; ramp rate: 9.42 Hz; MS averaging: 1; auto calibration: off. MS settings: scan: 100–1700 m/z; ion polarity: positive; scan mode: PASEF MS/MS number of PASEF ramps: 4; total cycle time: 0.5 s; charge minimum: 0 (unknown); charge maximum: 5; scheduling: target intensity: 12,500; intensity threshold: 1000. active exclusion: ON. The system sensitivity was controlled with 20 ng of HELA-digested proteins.

#### Peptide Identification

MSFragger searches were performed (via FragPipe) for the identification of non-tryptic peptides. Single databases were generated using Uniprot Microalgal proteins. Every sample was analyzed with the adequate database.

### 4.6. Bioactivity Assays

Bioactivity assays were performed on the 3 kDa permeates generated from both microalgal samples. Where enzyme inhibitory activity greater than 50% was observed for algal permeates, enzyme inhibitory IC_50_ values were also determined.

#### 4.6.1. ACE-1 Inhibition

The ACE-1 inhibitory assay was performed following the manufacturer’s instructions (Caymann Chemical Company, Ann Arbor, MI, USA). Samples were prepared in triplicate by re-suspending freeze-dried 3 kDa permeates in ddH_2_O to achieve a concentration of 1 mg/mL. Captopril© was used as the positive ACE-1 inhibitory control at a concentration of 1 mg/mL. Blank 1 and Blank 2 were prepared as the ACE-1 inhibitor control and reagent blank, respectively. The absorbance of each well was measured using a FLUOstar Omega microplate reader (BMG LABTECH GmbH, Offenburg, Germany) with a wavelength of 450 nm. Following the manufacturer’s instructions, ACE-1 inhibitory percentage values were calculated using the equation:$$ACE\;1\;inhibition\;\%=\frac{A_{blank\;1}\;-\;A_{sample}}{A_{blank\;1}-A_{blank\;2}}\;\times \;100$$

ACE-1 IC_50_ values for both algal permeate samples were determined by plotting the ACE-1 inhibition percentage values as a function of the sample concentration and solving the obtained function for 50% inhibition.

#### 4.6.2. ABTS Radical Scavenging Capacity

Samples were prepared in triplicate by suspending the microalgal 3 kDa permeate in DMSO to obtain a concentration of 1 mg/mL. A resveratrol/DMSO solution (1 mg/mL) was used as the positive control. The assay was performed in accordance with the manufacturer’s instructions (Caymann Chemical Company). Trolox (supplied in the kit) was used to plot the standard curve. The absorbance of each well was measured using a FLUOstar Omega microplate reader (BMG LABTECH GmbH, Offenburg, Germany) with a wavelength of 405 nm. The antioxidant concentration was calculated using the equation:$$Antioxidant\;\left[mM\right]\;=\;\frac{A_{sample}\;-\;q}{m}$$
where q and m refer to the y-intercept and the slope of the standard curve obtained using the Trolox standard, respectively. The antioxidant concentrations obtained were compared to the resveratrol antioxidant concentration values to obtain percentages of antioxidant activity values for the samples with respect to the positive control.

#### 4.6.3. α-Amylase Inhibition

The α-amylase inhibitory activity of the algal permeates was assessed using a colorimetric assay kit supplied by Abcam Ltd. (Abcam, Cambridge, UK). The assay was performed according to the manufacturer’s instructions. 1 mg/mL of each permeate sample was dissolved in assay buffer, and the absorbance of each well at 405 nm at time 0, 10, and 25 min was measured using a FLUOstar Omega microplate reader (BMG LABTECH GmbH, Offenburg, Germany). The slope of the kinetic profile obtained (using linear regression on three time points or assuming that the linear portion of the kinetic is between 0 and 10 min) was used to calculate the percentage relative α-amylase inhibition values using the formula:$$\%\;Relative\;inhibition_{\alpha -Amylase}\;=\;\frac{slope\;\left(EC\right)\;-\;slope\;\left(Sample\right)}{slope\;\left(EC\right)}\;\times \;100$$
where EC is the enzyme control.

#### 4.6.4. Cyclooxygenase Inhibitory Activity Assessment

Peptides IEWYGPDRPKFSPF and TVQIPGGERVPFLF, identified as being anti-inflammatory using the outlined in silico methodology, were chemically synthesized and tested in vitro for their potential anti-inflammatory/anti-pain activity using the Cyclooxygenase-2 (COX-2; EC 1.14.99.1) inhibition assay. Selected peptides were chemically synthesized by GenScript Biotech (Leiden, The Netherlands). GenScript also verified the purity of the peptide by analytical RP-HPLC–MS. Peptides were assessed for their ability to inhibit Cyclooxygenase 2 (COX-2) at a concentration of 1 mg/mL compared to a positive control (Diclofenc—a commercial drug, Merck, Dublin, Ireland) using the Cyclooxygenase-2 assay method according to the manufacturers’ instructions (Cayman Chemicals, Hamburg, Germany). Briefly, the synthesized peptides were incubated independently with human recombinant COX-2 (Cayman Chemicals, Hamburg, Germany). The assay was carried out in accordance with the manufacturer’s instructions. Both peptides inhibited COX-2 minimally—by 23.07% and 20.39%, respectively. 

### 4.7. Techno-Functional Activities

#### 4.7.1. Solubility

The protein solubility of the generated microalgae hydrolysates was determined using the protocol described by Beuchat and colleagues with some slight modifications [72]. The samples were prepared in quadruple by suspending the hydrolysates independently in ddH_2_O (1% *w/v*). The pH was adjusted to obtain pH values of 2, 4, 6, 8, and 10 using 1M HCl and 1M NaOH. Samples were subsequently shaken for 45 min and centrifuged for 30 min at 4000× *g*. The supernatants were collected, and the protein content of the suspended sample (C_tot_) and the supernatants (C_sup_) was measured using the QuantiPro BCA Assay kit (Sigma, St. Louis, MO, USA) as per the manufacturer’s instructions. The percentages of soluble proteins (S%) present in the hydrolysates were determined using the formula:$$S\%=\frac{C_{sup}}{C_{tot}}\times 100$$

#### 4.7.2. Water Holding (WHC) and Oil Holding Capacity (OHC)

The water holding capacity (WHC) and oil holding capacity (OHC) of hydrolysates were determined following the method described by Bencini [73]. Briefly, 1 g of dry microalgal hydrolysate was weighed and suspended in 10 mL of liquid (water for the WHC calculation and sunflower and rapeseed oil for OHC calculations). The samples were mixed for 1 min using a vortex mixer (Henry Troemner, Thorofare, NJ, USA) and then centrifuged at 2200× *g* for 30 min. The supernatant was decanted, and the remaining pellet was weighed. The WHC and the OHC were then calculated using the formula:$$WHC\backslash OHC=\frac{W_{2}-W_{1}}{W_{0}}\times 100=\frac{mass\;of\;retained\;liquid\;[g]}{mass\;of\;sample\;[g]}$$
where W_0_ is the mass of the dry sample, W_1_ is the mass of the dry sample in addition to that of the tube, and W_2_ is the mass of the tube plus the pellet.

#### 4.7.3. Emulsion Activity

The emulsion activity was evaluated using the method proposed by Naczk and colleagues [74] with slight modifications. A suspension of 0.01 g/mL of microalgal hydrolysate in ddH_2_O was prepared, and the pH was adjusted to obtain pH values of 2, 4, 6, 8, and 10 using 0.1M HCl and 0.1M NaOH. The samples were homogenized using a T25 Ultra-Turrax homogenizer (IKA^®^, Staufen im Breisgau, Germany) for 30 s at 14,000 rpm. Sunflower oil was used to create an emulsion (ratio of 3:2 oil/sample (v:w). The oil was added in two steps; after the first addition, the samples were homogenized for 30 s at 14,000 rpm, and then the rest of the oil was added and homogenized for one last time for 90 s at 14,000 rpm. The emulsion that formed was centrifuged at 1100× *g* 5 min. The emulsion activity (EA) was calculated as:$$EA\%=\frac{V_{E}}{V_{T}}\times 100$$
where V_E_ is the volume of the emulsion after the centrifugation step and V_T_ is the total volume of the sample inside the tube.

Emulsion heat stability (EHS) was determined by heating the emulsion previously prepared at 85 °C for 15 min and incubating at room temperature for 30 min, followed by centrifugation at 1100× *g* 5 min. The EHS was calculated as:$$EHS=\frac{V_{H}}{V_{O}}\times 100$$
where V_H_ is the volume of the emulsion after heating and V_O_ is the volume of the original emulsion.

### 4.8. Statistical Analysis

Experimental work was performed in triplicate, and results are expressed as the mean ± standard deviation (SD) of the replicates using EXCEL 2010. Measurements were performed in triplicate (*n* = 3) or (*n =* 9). Statistical analysis was conducted with Excel 2010. one-way ANOVA and a post hoc Tukey’s HSD test were applied—statistical significance was *p* < 0.05.

## 5. Conclusions

Novel hydrolysates with demonstrated in vitro bioactivities, as well as several new peptides generated from these *Chlorella* sp. and *Scenedesmus* sp. Hydrolysates, were identified, including peptides YDYIGNNPAKGGLF, YIGNNPAKGGLF, IEWYGPDRPKFL, RSPTGEIIFGGETM, TVQIPGGERVPFLF, and IEWYGPDRPKFLGPF. Combining hydrolysis of algae with two enzymes with MWCO filtration increased the content of protein. This approach also adds value to the resulting hydrolysates as it increases their bioactivities and potential health benefits. Hydrolysates generated demonstrate anti-ACE-1, anti-amylase, and antioxidant activities. They also have techno-functional properties, including emulsifying and OHC and WHC potential. This makes the hydrolysates suitable for use as ingredients to improve product functionality and health benefits. The identified bioactive peptides have predicted anti-inflammatory, anti-diabetic, and umami attributes. Such microalgal hydrolysates provide a protein source for consumers that has tertiary health benefits, which may improve human health. MS combined with in silico analysis and in vitro bioassays are useful tools for the drug discovery stage of novel compound isolation prior to animal and clinical testing.

## Figures and Tables

**Figure 1 marinedrugs-23-00053-f001:**
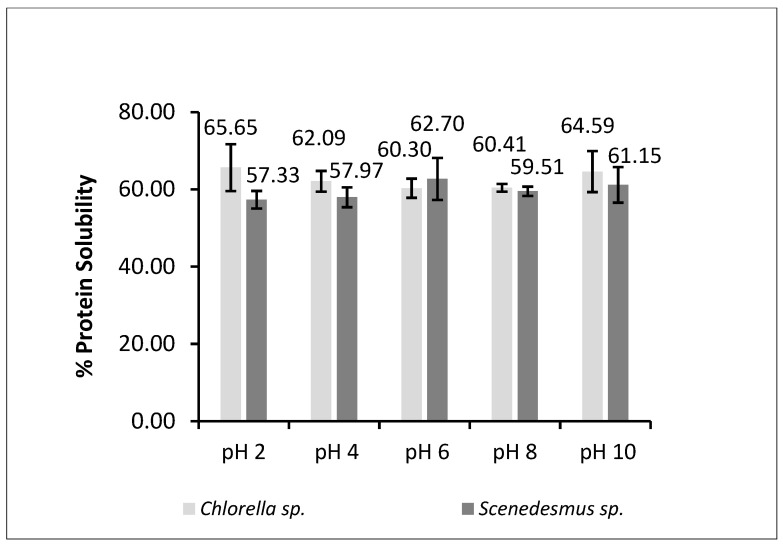
The percentage of soluble proteins found in microalgal hydrolysates generated using Viscozyme and Alcalase applied independently to *Chlorella* mix and *Scenedesmus* mix biomass.

**Figure 2 marinedrugs-23-00053-f002:**
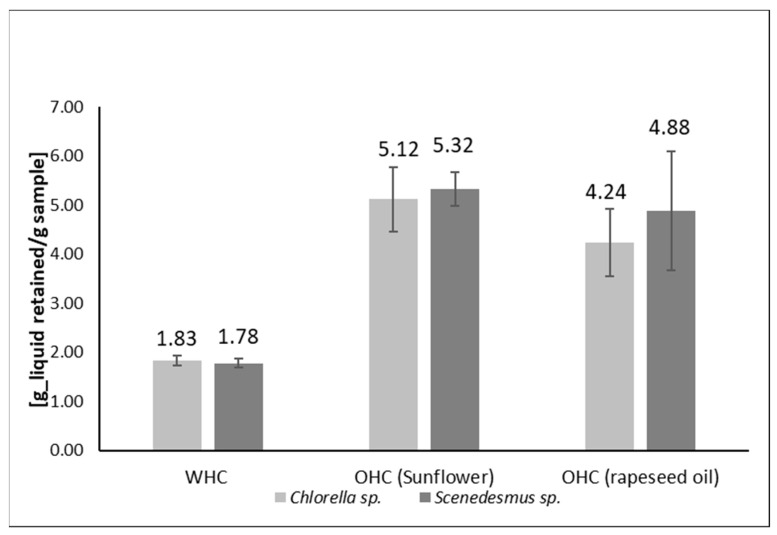
The WHC and OHC in sunflower and rapeseed oils of microalgae hydrolysates produced from *Chlorella* mix and *Scenedesmus* mix.

**Figure 3 marinedrugs-23-00053-f003:**
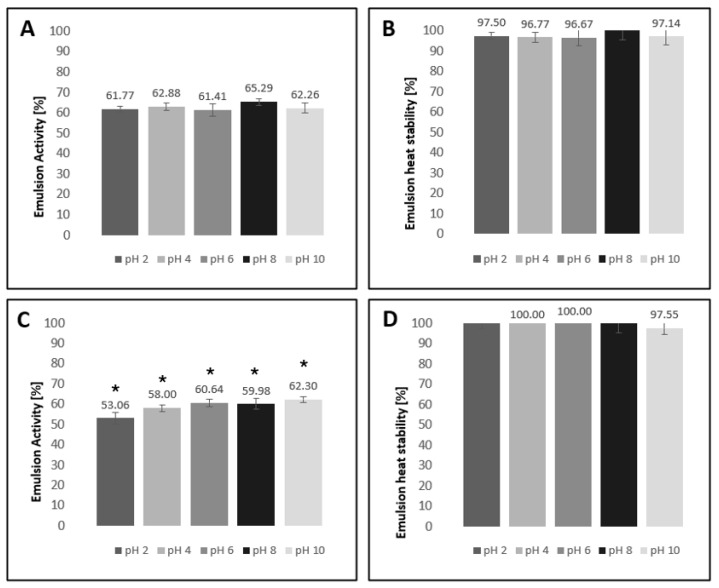
The emulsion activity (EA) and emulsion heat stability (EHS) of *Chlorella* mix (**A**,**B**) and *Scenedesmus* mix hydrolysate (**C**,**D**) were assessed at different pH levels. Columns with an (*) differ from each other by the Tukey test at a 5% probability.

**Figure 4 marinedrugs-23-00053-f004:**
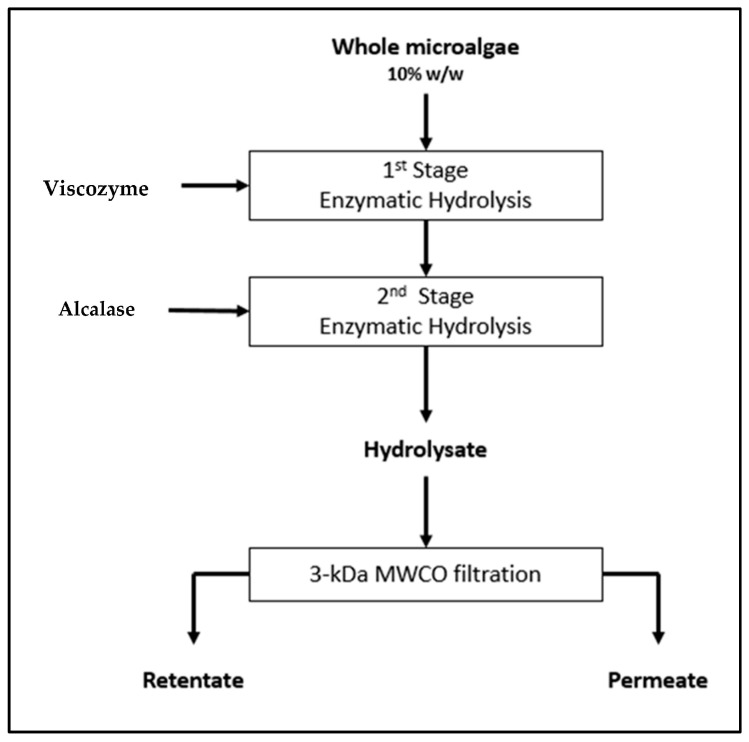
Process used to produce algal protein hydrolysates.

**Table 1 marinedrugs-23-00053-t001:** The degree of hydrolysis (DH) obtained by processing *Chlorella* mix and *Scenedesmus* mix with the selected enzymes.

Substrate	Enzyme	DH%
*Chlorella* mix	Viscozyme	25.41
*Chlorella* mix	Viscozyme + Alcalase	48.44
*Scenedesmus* mix	Viscozyme	27.46
*Scenedesmus* mix	Viscozyme + Alcalase	46.35

**Table 2 marinedrugs-23-00053-t002:** Proximate compositional analysis for whole algal biomass, produced hydrolysates, permeates, and retentates.

*Chlorella* Mix Fractions				
	*Chlorella* Mix Whole Biomass	*Chlorella* Mix Hydrolysate	*Chlorella* Mix Permeate	*Chlorella* Mix Retentate
Protein %	38.02	35.80	44.73	28.72
Fat %	13.95	13.61	0.65	12.05
Moisture %	7.93	8.85	16.16	8.16
Ash %	4.85	5.27	10.10	10.33
***Scenedesmus* mix fractions**				
	***Scenedesmus* mix whole biomass**	***Scenedesmus* mix hydrolysate**	***Scenedesmus* mix permeate**	***Scenedesmus* mix retentate**
Protein %	37.81	36.74	41.14	26.37
Fat %	10.16	6.32	1.07	5.58
Moisture %	12.96	12.48	16.93	31.35
Ash %	2.03	4.53	7.50	2.08

**Table 3 marinedrugs-23-00053-t003:** Summary of in vitro bioassay results obtained for *Chlorella* mix and *Scenedesmus* mix hydrolysate 3 kDa permeates.

Sample	ACE-1 Inhibition (%)	Antioxidant Activity (%)	Relative α-Amylase Inhibition (%)
*Chlorella* mix 3 kDa Permeate	88.07 ± 1.69	72.54 ± 18.16	71.32 ± 12.30
*Scenedesmus* mix 3 kDa Permeate	86.24 ± 2.89	76.17 ± 8.92	28.78 ± 0.77

**Table 4 marinedrugs-23-00053-t004:** Peptide sequences identified from *Chlorella* mix and *Scenedesmus* mix hydrolysate permeate fractions generated using Viscozyme, Alcalase, and bioactivities of peptide fragments obtained after simulated gastrointestinal digestion.

Peptide Sequence	Protein of Origin	Peptide Ranker	BIOPEP-UWM Search	PreAIP	Umami-MRNN	Anti-DMP-Pred	Microalgal Biomass Origin
YDYIGNNPAKGGLF (99%)	Photosystem II CP47 reaction center protein OS = Pdinomonas minor OX = 3159 GN = pbB PE = 3 SV = 1;	0.735	Novel	0.428 (medium confidence anti-inflammatory peptide (AIP))	non-umami	0.49 (Not an anti-diabetic peptide)	*Chlorella* sp. mix-derived peptide
YIGNNPAKGGLF (95%)	Photosystem II CP47 reaction center protein OS = Pdinomonas minor OX = 3159 GN = pbB PE = 3 SV = 1;	0.806	Novel	0.374 (low confidence AIP)	umami, predicted threshold: 29.685139 mmol/L	0.46 (Not an anti-diabetic peptide)	*Chlorella* sp. mix-derived peptide
IEWYGPDRPKFL (99%)	Chlorophyll a-b binding protein, chloroplastic OS = Chlamydomonas reinhardtii OX = 3055 GN = LhcII-3 PE = 1 V = 1	0.811	Novel	0.627 (High confidence AIP)	non-umami	0.33 (Not an anti-diabetic peptide)	*Scenedesmus* sp. mix-derived peptide
RSPTGEIIFGGETM (99%)	Photosystem II CP47 reaction center protein OS = Pdinomonas minor OX = 3159 GN = pbB PE = 3 SV = 1;	0.258	Novel	0.451 (Medium confidence AIP)	umami, predicted threshold: 10.636716 mmol/L	0.6 (likely to have anti-diabetic properties)	*Scenedesmus* sp. mix-derived peptide
TVQIPGGERVPFLF (99%)	Oxygen-evolving enhancer (Fragment) OS = Cyanidioschyzon merolae OX = 45,157 GN = pbO PE = 3 SV = 1	0.592	Novel	0.522 (High confidence AIP)	umami, predicted threshold: 19.447886 mmol/L	0.57 (likely to have anti-diabetic properties)	*Scenedesmus* sp. mix-derived peptide
IEWYGPDRPKFLGPF (99%)	Chlorophyll a-b binding protein, chloroplastic OS = Chlamydomonas reinhardtii OX = 3055 GN = LhcII-3 PE = 1 V = 1	0.904	Novel	0.472 (High confidence AIP)	non-umami	0.43 (Not an anti-diabetic peptide)	*Scenedesmus* sp. mix-derived peptide

**Table 5 marinedrugs-23-00053-t005:** Details of the microalgal mixtures, their composition and delivery format.

Algal Mixture Name	Composition	(%)	Form Delivered
Scenedesmus mixture	*Scenedesmus* sp.	80	Spray-dried
	Diatomee	17	
	*Chlorella* sp.	3	
Chlorella mixture	*Chlorella* sp.	92.5	Spray-dried
	*Scenedesmus* sp.	5	
	*Chlorococcum*	2.5	

## Data Availability

The generated data are available from the corresponding author and all data are presented in this paper.

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
