# Peer review of "Functional and Bioactive Benefits of Selected Microalgal Hydrolysates Assessed In Silico and In Vitro"

_marinedrugs, 2025, doi:10.3390/md23020053_

Round 1
Reviewer 1 Report
Comments and Suggestions for Authors
The authors have prepared a manuscript devoted to the biotechnological processing of microalgae. The article is written in a classical manner and at a good scientific level for the relevant journal. However, I believe that it needs significant revision for publication in the journal Marine Drugs.
Firstly, the most important point is the discovery of hydrolysate peptides that may be useful in medicine. These peptides, detected by MS analysis, are part of important functional proteins and, if they are blast analyzed in the NCBI database, are not new and/or unique. Unfortunately, it is impossible to evaluate the MS method used, since there is no description of this method in the corresponding section. In addition, there is no appendix or supplementary file where one could see other peptides found to evaluate their potential.
Secondly, the attempt to synthesize two peptides and study their potential is described inappropriately - there is no method for either their synthesis or determination of inhibitory activity to COX-2. A description of this result was inserted into the MS and in silico analysis section (?).
Thirdly, as it is clear from the materials and methods, there were two mixtures of microalgae, conventionally named by the authors according to the predominant species... However, the names of Scenedesmus sp. and Chlorella sp. cannot be applied to the mixtures, this looks incorrect. There is no information how and by whom the species identity of microalgae was determined.
Minor comments include negligence in the design of the article:
Line 73. and Tetraselmis sp. should be written as “and Tetraselmis sp.”
Lines 122, 146 At the end of sentences there should be references to cited sources.
Lines 126,236 in vitro should be written as in vitro
Lines 131,146, 251,396 sp. should be written as sp.
Lines 233-235 The content is no relevant to Section name
Line 237 m/mL must be fixed
Line 246 Combining two activities in one subsection is not logically justified. Should be separated.
Lines 247, 257 were should be replaced was
Line 250 76,17% should be written as 76.17%
Line 256 S. obliquus should be written as S. obliquus
Line 396 diatoms should be written as diatoms
Line 567 References are not formatted according to the journal rules
The lack of novelty and the impossibility of evaluating the obtained results do not allow us to give a positive recommendation for publication.
Author Response
Marine Drugs-3398639
Response to reviewer comments (Reviewer 1):
Comments and Suggestions for Authors
1. Comment: The authors have prepared a manuscript devoted to the biotechnological processing of microalgae. The article is written in a classical manner and at a good scientific level for the relevant journal. However, I believe that it needs significant revision for publication in the journal Marine Drugs.
1. Response: We would like to thank reviewer 1 for their comments. We address these comments in the following response to author comments.
2. Comment: Firstly, the most important point is the discovery of hydrolysate peptides that may be useful in medicine. These peptides, detected by MS analysis, are part of important functional proteins and, if they are blast analyzed in the NCBI database, are not new and/or unique.
2. Response: Peptides are sequences of between 2-30 amino acids in length. The aim of our work was to identify bioactive peptides. These are peptide sequences between 2-30 amino acids in length that have to be cut from the parent protein to have bioactivities and potential health benefits. While the sequences of peptides, if put into NCBI BLAST are not novel, the identification of the exact amino acid sequence that has a bioactivity is novel. Several published papers exist, including many written by our group that follow this methodology to identify novel, bioactive peptides. We disagree with reviewer 1 here that the peptides are not novel. The identification of the peptide sequence is novel and using in silico analysis to determine novelty is novel. Several peptides identified were not bioactive in this work and we highlight this in the table. The proteins from which the peptides are derived are not novel and are characterised but the peptides, cut from the parent protein using hydrolysates are novel in terms of their bioactivities and the fact that these sequences have bioactivities. The peptides are not reported in the literature or further in databases hosting bioactive peptides like BIOPEP.
3. Comment: Unfortunately, it is impossible to evaluate the MS method used, since there is no description of this method in the corresponding section. In addition, there is no appendix or supplementary file where one could see other peptides found to evaluate their potential.
3. Response: We agree with the reviewer and we have added the following text to the MS methods section: “4.5 Mass spectrometry in tandem analysis
1 µL of every sample were quantified by nanoDrop (A 280 nm, e= 1mg/mL). For every sample, the volume corresponding to 190 ng was diluted in 20 µL with 0.1% FA and loaded in an Evotip pure tip (EvoSep) according to the manufacturer instructions. Tandem mass spectrometry analysis (LC–MS/MS) was performed in a Tims TOF fleX mass spectrometer (Bruker). The sample loaded in the Evotip pure tip was eluted to an analytical column (EvoSep 15 cm x 150 µm, 1.5 µm; Evosep) by the Evosep One system, and solved with the 30 SPD chromatographic method defined by the manufacturer. Eluted peptides were ionized in a captive Spray with 1700 V at 200 ºC, and analyzed using ddaPASEF mode with the following settings: TIMS settings Mode: custom; 1/K0: 0.7-1.76 V.s/cm2; ramp time: 100 ms; Duty Cycle: 100%; Ramp Rate: 9.42 Hz; Ms Averaging:1; Auto Calibration: off. MS settings Scan: 100-1700 m/z; Ion Polarity: Positive; Scan Mode: PASEF MS/MS Number of PASEF ramps: 4; Total Cycle time: 0.5 s; Charge Minimun:0 (unknown); Charge maximum: 5;Scheduling: target Intensity: 12500, Intensity Threshold: 1000. Active exclusion: ON. The system sensitivity was controlled with 20 ng of HELA digested proteins.
4.5.1 Peptide identification
MSFragger searches were performed (via FragPipe) for the identification of non-tryptic peptides. Single databases were generated using Uniprot Microalgal proteins. Every sample was analyzed with the adequate database.”
4 Comment: Secondly, the attempt to synthesize two peptides and study their potential is described inappropriately - there is no method for either their synthesis or determination of inhibitory activity to COX-2. A description of this result was inserted into the MS and in silico analysis section (?).
4 Response: We have included a description for the synthesis of peptides in the Materials and Methods section and additionally the method used to determine the inhibitory activity to COX-2. Please see the following text:
“4.6.4 Cyclooxygenase inhibitory activity assessment
Peptides IEWYGPDRPKFSPF and TVQIPGGERVPFLF identified as being anti-inflammatory using the in silico methodology outlined were chemically synthesized and tested in vitro for their potential anti-inflammatory/anti-pain activity using the Cyclooxygenase-2 (COX-2; EC 1.14.99.1) inhibition assay. Selected peptides were chemically synthesised by GenScript Biotech (Leiden, The Netherlands). GenScript also verified the purity of the peptide by analytical RP-HPLC–MS. Peptides were assessed for their ability to inhibit Cyclooxygenase 2 (COX-2) at a concentration of 1 mg/mL compared to a positive control (Diclofenc – a commercial drug, Merck, Dublin, Ireland) using the Cyclooxygenase-2 assay method according to the manufacturers’ instructions (Cayman Chemicals, Hamburg, Germany). Briefly, the synthesised peptides were incubated independently with human recombinant COX-2 (Cayman Chemicals, Hamburg, Germany). The assay was carried out in accordance with the manufacturer’s instructions. Both peptides inhibited COX-2 minimally – by 23.07 % and 20.39% respectively.”
5 Comment: Thirdly, as it is clear from the materials and methods, there were two mixtures of microalgae, conventionally named by the authors according to the predominant species... However, the names of Scenedesmus sp. and Chlorella sp. cannot be applied to the mixtures, this looks incorrect. There is no information how and by whom the species identity of microalgae was determined.
5 Response: We agree with the reviewer. We have described in more detail the algal mixtures and how they were characterised by partners FZJ Julich Germany. We have included the following text in the Materials and Methods:“4.1. Biomass
FZJ Jülich, Germany as part of the IDEA project, supplied two mixed algal biomass samples to Teagasc. The first sample, harvested in June 2018, called Chlorella mix, was composed of Chlorella sp. (92.5%, w/w), Scenedesmus sp. (5%, w/w) and Chlorocococcus sp. (2.5% w/w). The second sample, harvested in May 2018, termed Scenedesmus mix, is mainly composed of Scenedesmus sp. and included Chlorella sp. and other diatoms (Table 5). All biomass was provided in a spray-dried powder form. The composition of each sample was determined by microscopy by Bärbel Ackermann (FZJ Julich, Germnay.
Table 5: Details of the microalgal mixtures, their composition and delivery format.
|
Algal mixture name |
Composition |
(%) |
Form delivered |
|
Scenedesmus mixture |
Scenedesmus sp. |
80 |
Spray-dried |
|
Diatomee |
17 |
||
|
Chlorella sp. |
3 |
||
|
Chlorella mixture |
Chlorella sp. |
92.5 |
Spray-dried |
|
Scenedesmus sp. |
5 |
||
|
Chlorococcum |
2.5 |
Minor comments include negligence in the design of the article:
- Comment: Line 73. and Tetraselmis sp. should be written as “and Tetraselmis sp.” Response: We have corrected this to sp.
- Comment: Lines 122, 146 At the end of sentences there should be references to cited sources. Response: We have included references at the end of the lines as requested.
- Comment: Lines 126,236 in vitro should be written as in vitro Response: We have written it in vitro
- Comment: Lines 131,146, 251,396 sp.should be written as sp. Response: We have corrected all to sp.
10 Comment: Line 237 m/mL must be fixed
10 Response: We have corrected this in the text to mg/mL.
11 Comment: Line 246 Combining two activities in one subsection is not logically justified. Should be separated.
11 Response: We have separated the two sections. It now reads as follows:
“2.3.2. ABTS Radical Scavenging Effect
The antioxidant effect of generated microalgal-3-kDa permeates were assessed using the ABST scavenging assay. When both algal permeates were assessed at 1 mg/mL concentrations using the positive control Resveratrol the ABTS-antioxidant activity percentage values obtained were 72.54 % and 76,17% for the 3-kDa permeates generated from the Chlorella sp. and Scenedesmus sp. hydrolysates, respectively (Table 3). The antioxidant activities of microalgal hydrolysates were reported previously. Alzahrani and colleagues assessed the antioxidant activity of Chlorella vulgaris biomass hydrolysed with Alcalase previously. This study identified an ABTS radical scavenging activity value of 63% [53]. Additionally, Afify and colleagues report an ABTS radical scavenging activity for S. obliquus hydrolysed with Papain of 87.03% [54].
2.3.3 α-amylase inhibitory activity
The α-amylase inhibitory potential of generated algal 3-kDa permeates were also assessed and a 71.32 ± 12.30 % value was obtained for the Chlorella sp. 3-kDa permeate fraction. Table 3 collates the bioactivity results obtained for the 3-kDa permeate fractions generated from Chlorella sp. and Scenedesmus sp. Viscozyme and Alcalase hydrolysates, respectively.”
12 Comment: Lines 247, 257 were should be replaced was
12 Response: We have changed were to was as follows: “The antioxidant activities of microalgal hydrolysates was reported previously.” And “The α-amylase inhibitory potential of generated algal 3-kDa permeates was also assessed and a 71.32 ± 12.30 % value was obtained for the Chlorella sp.”.
13 Comment: Line 250 76,17% should be written as 76.17%
13 Response: We have corrected 76,17% to 76.17%.
14 Comment: Line 256 S. obliquus should be written as S. obliquus
14 Response: We have corrected this text to read S. obliquus.
15 Comment: Line 396 diatoms should be written as diatoms
15 Response: We have edited this to read diatoms.
The lack of novelty and the impossibility of evaluating the obtained results do not allow us to give a positive recommendation for publication.

Reviewer 2 Report
Comments and Suggestions for Authors
Comments
The presented work is related to the technical and functional analysis of the properties of extracts of a number of algae, and is of interest to developers of new trends in algae processing.
There are comments on the article:
1. In the text of the article, the authors repeatedly state that the peptides isolated from algae and discussed in this work are recommended for use in food products. To isolate the peptides, the authors use proteolysis with Viscozyme. It is well known that many proteases, including Viscozyme, have a bitter taste - this is not at all desirable in food products. Question: why was no organoleptic analysis of the taste of the peptides carried out in the work?
2. When isolating the peptides, the samples were subjected to thermal deactivation in a water bath at 80 °C. At this temperature, peptide denaturation already occurs. Question: how did the authors take this into account when interpreting the results?
3. The article does not describe how the statistical processing of the experimental data was carried out.
4. In the conclusions, the authors state that the peptides studied in the work are suitable ingredients for use as an alternative to existing emulsifiers, such as emulsifiers derived from soy. To make such a statement, the article must present comparative data on the properties of soy proteins as emulsifiers. There is no such data. In this regard, in my opinion, the statement is incorrect.
5. The conclusions to the work are built from general phrases that are little related to the purpose of the work. The introduction states: The purpose of this work is to obtain bioactive peptides containing protein hydrolysates from selected microalgae Chlorella sp. and Scenedesmus sp. using a two-stage hydrolysis process, and to evaluate the technological and functional properties of the obtained hydrolysates. In my opinion, the conclusions should be changed and built on specific data in accordance with the results obtained.

Author Response
Reviewer comments
Comment 1, reviewer 1:
In the text of the article, the authors repeatedly state that the peptides isolated from algae and discussed in this work are recommended for use in food products. To isolate the peptides, the authors use proteolysis with Viscozyme. It is well known that many proteases, including Viscozyme, have a bitter taste - this is not at all desirable in food products. Question: why was no organoleptic analysis of the taste of the peptides carried out in the work?
Response: We would like to thank the reviewer for this intelligent comment. It is true that we would see the application of these peptide containing hydrolysates and peptides in food or functional food products. It is true that Viscozyme can cause a bitter taste and it is ordinarily used on carbohydrates. We have combined it in this instance with Alcalase. We didn’t focus on taste analysis as our focus was on generation of bioactive peptides with health benefits and a study on the organoleptic analysis of peptides was beyond the scope of our funding as we would be required to carry out a large scale taste trial with synthesized peptides. To manufacture the peptide in mg quantities (20 mg) costs thousands of euros and for a taste trial to determine the organoleptic properties it would cost a lot more so this was beyond our scope. We did however carry out in silico umami analysis of the peptides and several of them cause umami flavour which is a desirable attribute for food products and we have reported on this within the text of the article in Table 4, Line 294-297. In the revision, we have also used the programme Umami YYDS (tastepeptides-meta.com) (as well as Umami-MRNN- used previously and described in the original text) to assess the umami and bitterness of the peptides listed from the algal hydrolysates in Table 4 and report on these in the results as well, lines 288-294 “In addition, the umami and bitterness of the peptides identified from algae and listed in Table 4 was assessed using Umami_YYDS 2.0 BETA an Umami/Bitterness Judgment Model Based on Machine Learning and Chemical Descriptors. The peptide YDYIGNNPAKGGLF, classifier is Umami and the probability is 1.0. The peptide IEWYGPDRPKFL, classifier is Umami and the probability is also 1.0 and the peptide TVQIPGGERVPFLF, classifier is Umami, with the probability being 0.8326 (accessed on 30th December 2024).”
Comment 2, reviewer 1:
When isolating the peptides, the samples were subjected to thermal deactivation in a water bath at 80 °C. At this temperature, peptide denaturation already occurs. Question: how did the authors take this into account when interpreting the results?
Response:
We have described the steps used to generate the hydrolysate including the enzyme deactivation step in the manuscript. We could have run a control without enzyme to see if the temperature releases peptides but we did not do this as it was not necessary. The steps we describe, combining two enzyme and enzyme deactivation steps using temperature are what generated the results we are reporting on in this paper. We have used similar methodologies previously, with different enzymes and generate different peptides. It is the enzyme that is releasing the peptides from the substrate material, not the temperature used to deactivate the enzyme. If it was temperature, we would get the same peptide consistently from the same biomass.
Comment 3, reviewer 1:
The article does not describe how the statistical processing of the experimental data was carried out.
Response: We thank the reviewer for this comment. We have included how the statistics were processed now in the revised manuscript. See lines “Experimental work was performed in triplicate, and results are expressed as the mean ± standard deviation (SD) of the replicates using EXCEL 2010. Measurements were performed in triplicate (n = 3) or (n = 9). Statistical analysis was conducted with Excel 2010. one-way ANOVA and a post hoc Tukey’s HSD test was applied - statistical significance was p < 0.05.", lines 550-556.
Comment 4, reviewer 1:
In the conclusions, the authors state that the peptides studied in the work are suitable ingredients for use as an alternative to existing emulsifiers, such as emulsifiers derived from soy. To make such a statement, the article must present comparative data on the properties of soy proteins as emulsifiers. There is no such data. In this regard, in my opinion, the statement is incorrect.
Response: We have edited the conclusion and we have removed the comparison with soy protein as we only have referenced data on this and not direct comparative data. We hope the revised text fulfils the request of reviewer 1.
Comment 5, reviewer 1:
The conclusions to the work are built from general phrases that are little related to the purpose of the work. The introduction states: The purpose of this work is to obtain bioactive peptides containing protein hydrolysates from selected microalgae Chlorella sp. and Scenedesmus sp. using a two-stage hydrolysis process, and to evaluate the technological and functional properties of the obtained hydrolysates. In my opinion, the conclusions should be changed and built on specific data in accordance with the results obtained.
Response: We agree with the reviewer and we have edited the text of the conclusion section to reflect the aims and introduction. The conclusion section now reads as follows “Novel hydrolysates with demonstrated in vitro bioactivities as well as several new peptides generated from these Chlorella sp. and Scenedesmus sp. hydrolysates were identified including peptides YDYIGNNPAKGGLF, YIGNNPAKGGLF, IEWYGPDRPKFL, RSPTGEIIFGGETM, TVQIPGGERVPFLF and IEWYGPDRPKFLGPF. Combining hydrolysis of algae with two enzymes with MWCO filtration increased the content of protein. This approach also adds value to the resulting hydrolysates as it increases their bioactivities and potential health benefits. Hydrolysates generated have demonstrated anti-ACE-1, anti-amylase and antioxidant activities as well as techno-functional properties including emulsifying and OHC and WHC potential. This makes the hydrolysates suitable ingredients for use as ingredients to improve product functionality and health benefits. The identified bioactive peptides have predicted anti-inflammatory, anti-diabetic and umami attributes. Such microalgal hydrolysates could provide essential amino acids to consumers as well as tertiary, health benefits to improve human global health. MS combined with in silico analysis and in vitro bioassays are useful tools in drug discovery and production of marine drugs.”

Reviewer 3 Report
Comments and Suggestions for Authors
The presented article is devoted to the production of bioactive peptides from microalgae Chlorella sp. and Scenedesmus sp. using enzymatic hydrolysis. In addition to the analysis of physiological activity and functional properties of the entire hydrolysate, the authors use mass-spectral identification of peptides, which allows for a more in-depth study of the biologically active properties of the hydrolysate and its potential application.
The following remarks need to be corrected:
- The phrase concerning the UN should be excluded from the abstract, since it should contain information about the scientific work performed. This phrase should be moved to the introduction. Instead of this phrase, it should be indicated that the enzymatic hydrolysis was carried out in two stages and the names of the enzyme preparations used should be given.
- In Table 1, the column labeled "sample" can be divided into two columns, "substrate" and "enzyme".
- Fig1 and Fig2. These figures show the mean values ​​and indicate the ranges of standard deviations. It is noticeable that the standard deviations are comparable with the difference in the mean values ​​obtained for two substrates. It is necessary to indicate with letters that the mean values ​​are statistically different. For Figure 3, it seems that the values ​​do not depend on pH (maybe except for Fig.3 c). A statistical significance test will provide clarity.
- The experimental part lacks mass-spectrometry details. It is necessary to add at least a brief description.
Author Response
Response to reviewer comments
Reviewer 3: marinedrugs-3398639
Reviewer comments
Open Review
(x) I would not like to sign my review report
( ) I would like to sign my review report
Quality of English Language
(x) The quality of English does not limit my understanding of the research.
( ) The English could be improved to more clearly express the research
|
Yes |
Can be improved |
Must be improved |
Not applicable |
|
|
Does the introduction provide sufficient background and include all relevant references? |
(x) |
( ) |
( ) |
( ) |
|
Is the research design appropriate? |
(x) |
( ) |
( ) |
( ) |
|
Are the methods adequately described? |
( ) |
(x) |
( ) |
( ) |
|
Are the results clearly presented? |
( ) |
(x) |
( ) |
( ) |
|
Are the conclusions supported by the results? |
(x) |
( ) |
( ) |
( ) |
Comments and Suggestions for Authors
Comment 1: The presented article is devoted to the production of bioactive peptides from microalgae Chlorella sp. and Scenedesmus sp. using enzymatic hydrolysis. In addition to the analysis of physiological activity and functional properties of the entire hydrolysate, the authors use mass-spectral identification of peptides, which allows for a more in-depth study of the biologically active properties of the hydrolysate and its potential application.
The following remarks need to be corrected:
- The phrase concerning the UN should be excluded from the abstract, since it should contain information about the scientific work performed. This phrase should be moved to the introduction. Instead of this phrase, it should be indicated that the enzymatic hydrolysis was carried out in two stages and the names of the enzyme preparations used should be given.
Response 1: We thank the reviewer for their comments and we have edited the text accordingly. We have removed the phrase concerning the UN from the introduction and we have indicated that the enzymatic hydrolysis was carried out in two stages using the enzymes Viscozyme and Alcalase. Please see the following text in the introduction:
“BIOPEP-UWM, a peptide database, contains 5128 peptides from a myriad of resources. Five listed peptides are Angiotensin-I-converting enzyme (ACE-1; EC3.4.15.1) inhibitory peptides derived from a red alga, while two from Chlorella vulgaris have anti-cancer and antioxidative bioactivities. Herein, we describe a process combining hydrolysis with two enzymes Alcalase and Viscozyme, and filtration to generate protein-rich, bioactive peptide-containing hydrolysates from mixed species of Chlorella sp. and Scenedesmus sp. The potential of generated algal hydrolysates to act as food ingredients was determined by assessment of their techno-functional (foaming, emulsification, solubility, water holding and oil holding capacity) properties. Bioactive screening of hydrolysates in vitro combined with mass spectrometry (MS), and in silico, predictions identified bioactive and functional hydrolysates and six novel peptides. Peptides derived from Chlorella sp. have the sequences YDYIGNNPAKGGLF and YIGNNPAKGGLF with predicted anti-inflammatory (medium confidence) and umami potential. Peptides from Scenedesmus sp. have sequences IEWYGPDRPKFL, RSPTGEIIFGGETM, TVQIPGGERVPFLF and IEWYGPDRPKFLGPF with predicted anti-inflammatory, anti-diabetic and umami attributes. Such microalgal hydrolysates could provide essential amino acids to consumers as well as tertiary, health benefits to improve human global health.”
Comment 2: - In Table 1, the column labeled "sample" can be divided into two columns, "substrate" and "enzyme".
Response 2: We have divided Table 1 and the column labelled sample into two columns labelled substrate and enzyme.
Table 1. The Degree of hydrolysis (DH) obtained by processing Chlorella sp. and Scenedesmus sp. with the selected enzymes.
|
Substrate |
Enzyme |
DH% |
|
Chlorella sp. |
Viscozyme |
25.41 |
|
Chlorella sp. |
Viscozyme + Alcalase |
48.44 |
|
Scenedesmus sp. |
Viscozyme |
27.46 |
|
Scenedesmus sp. |
Viscozyme + Alcalase |
46.35 |
Comment 3: - Fig1 and Fig2. These figures show the mean values ​​and indicate the ranges of standard deviations. It is noticeable that the standard deviations are comparable with the difference in the mean values ​​obtained for two substrates. It is necessary to indicate with letters that the mean values ​​are statistically different. For Figure 3, it seems that the values ​​do not depend on pH (maybe except for Fig.3 c). A statistical significance test will provide clarity.
Response 3: We have taken into account the reviewers comments and we have added data regarding the statistical analysis of data. See lines “Experimental work was performed in triplicate, and results are expressed as the mean ± standard deviation (SD) of the replicates using EXCEL 2010. Measurements were performed in triplicate (n = 3) or (n = 9). Statistical analysis was conducted with Excel 2010. one-way ANOVA and a post hoc Tukey’s HSD test was applied - statistical significance was p < 0.05.", lines 550-556.”
Figure 1 – There are no statistical differences observed either between samples at different pH values nor between different biomass or between the same biomass at different pH values. Considering that, we updated the paragraph 2.2.1.
Figure 2 see lines “There is no statistical difference between the OHC of the two studied biomass hydrolysates and data does not show a significant difference between the OHC when different oils were used (Figure 2).”
In Figure 3 (*) has been added to address the statistical significance of the results. The label now reads:” Figure 3. Emulsion activity (EA) and emulsion heat stability (EHS) of Chlorella sp. (A, B) and Scenedesmus sp. hydrolysate (C, D) assessed at different pH values. Column with (*) differ from each other by Tukey test at 5% probability.”
Comment 4: The experimental part lacks mass-spectrometry details. It is necessary to add at least a brief description.
Response 4: 4.5 Mass spectrometry in tandem analysis
1 µL of every sample were quantified by nanoDrop (A 280 nm, e= 1mg/mL). For every sample, the volume corresponding to 190 ng was diluted in 20 µL with 0.1% FA and loaded in an Evotip pure tip (EvoSep) according to the manufacturer instructions. Tandem mass spectrometry analysis (LC–MS/MS) was performed in a Tims TOF fleX mass spectrometer (Bruker). The sample loaded in the Evotip pure tip was eluted to an analytical column (EvoSep 15 cm x 150 µm, 1.5 µm; Evosep) by the Evosep One system, and solved with the 30 SPD chromatographic method defined by the manufacturer. Eluted peptides were ionized in a captive Spray with 1700 V at 200 ºC, and analyzed using ddaPASEF mode with the following settings: TIMS settings Mode: custom; 1/K0: 0.7-1.76 V.s/cm2; ramp time: 100 ms; Duty Cycle: 100%; Ramp Rate: 9.42 Hz; Ms Averaging:1; Auto Calibration: off. MS settings Scan: 100-1700 m/z; Ion Polarity: Positive; Scan Mode: PASEF MS/MS Number of PASEF ramps: 4; Total Cycle time: 0.5 s; Charge Minimun:0 (unknown); Charge maximum: 5;Scheduling: target Intensity: 12500, Intensity Threshold: 1000. Active exclusion: ON. The system sensitivity was controlled with 20 ng of HELA digested proteins.
4.5.1 Peptide identification
MSFragger searches were performed (via FragPipe) for the identification of non-tryptic peptides. Single databases were generated using Uniprot Microalgal proteins. Every sample was analyzed with the adequate database.

Round 2
Reviewer 1 Report
Comments and Suggestions for Authors
Accept in present form
Author Response
Response to editor comments Marine Drugs Paper _ 3398639
16th January 2025
Editor comments: The manuscript was improved but I would suggest some modification to improve the work. The authors identify new peptides obtained from a double process of enzymatic treatment and subsequent membrane filtration from two species of microalgae/mix and they characterised their bioactive functions.
Comment 1: Please add in MM more information on membrane filtration parameter: Type of membrane, flow rate and more.
Response 1: We have taken on board the editors comments and we have included the following information in the paper concerning the membrane size and filtration rate. On line 402 we mention the membrane size (3kDa). In the M&M section, we have included the following text: “Hydrolysates were filtered through a 3-kDa cellulose cartridge filter (Merck Millipore, Cork, Ireland) with a molecular weight cut off (MWCO) of 3-kDa to generate a permeate fraction. The 3-kDa Millipore Prep/Scale Tangible Flow Filtration (TFF) system (Merck Millipore, Cork, Ireland) was employed for this purpose. Deionized water (Nano-water purification and filtration system, Merck, Dublin, Ireland) was used to flush the permeate from the system at a flow rate of 10 mL/minute.”
Comment 2: Discussion of results is redundant with the results paragraph. I ask the authors to make an effort to discuss the results, which, will be useful to further highlight the strengths of the work.
Response 2: We agree with the reviewer and we have changed the text as follow (Introduction, Results) to highlight further the strengths of the work:
Introduction: We have inserted the following sentence to highlight the benefits of the described work in Materials and Methods and the Results section “To offset processing costs, it is important to expand the market value of hydrolysates. This is possible by identifying the tertiary benefits (health and functional) of the peptides found in end process hydrolysates.”
Results section: Results and their significance are discussed as follows:
In relation to protein solubility we have inserted the following text: “Microalgae proteins examined in other studies show low solubility at pH < 5 and solubility increases from neutral to alkaline pH, as is the case in our work. Protein solubility controls gelation, foaming, and emulsifying capacity and is a measure of food quality in relation to sedimentation, viscosity and turbidity. Results in Figure 1 suggest that the hydrolysates have a broad solubility range and could be a good fit for food formulations at different pH values.”
&
The identified OHC values are discussed, lines X-X as follows: “Oil-holding capacity (OHC) refers to the ability of a protein or hydrolysate to bind with lipids and influences sensory characteristics of foods, in particular, the mouthfeel and flavor of foods. The OHC values identified for Scenedesmus sp. mix and Chlorella sp. mix are comparable to chickpea and cottonseed protein isolates, but are less than values reported for yellow pea and chickpea protein concentrates made using isoelectric precipitation (50, 51).” A new reference from 2024 was also inserted in the reference list.
&
For the observed results concerning emulsification, we inserted the following text: “Both hydrolysates demonstrate optimum emulsion activities at pH 10 and emulsion stabilities at pH 10 and 6, for Chlorella mix hydrolysate and Scenedesmus mix hydrolysate, respectively. In other work, the best emulsifying capacity of Chlorella vulgaris proteins was obtained at pH 7 (50). The pH value for which emulsion activity is optimal determines the use of algae in food products (50). Emulsions that are stable at neutral or alkaline pH values, as is the case here, are suited to neutral/basic food preparations, usually preparations like alkaline drinks containing fruit or vegetables or nut “dairy alternative” drinks.”
& In relation to bioactive studies, we have included the following text:
“The ACE-1 inhibitory values obtained for both hydrolysates combined with the identified peptides in Table 4 highlight the potential of these hydrolysates to have anti-hypertensive activities as they compare favorably to Captopril© the antihypertensive drug in terms of inhibition of ACE-1. Further in vivo trials, in Spontaneously Hypertensive Rats (SHRs) would confirm an antihypertensive effect.”
& In relation to the alpha amylase inhibition results, we include the following sentence “Use of these microalgal hydrolysates is a potential strategy & therapeutic approach to reduce blood glucose levels in the management of T2D.”
& In relation to the MS discussion of results and in silico discussion of results we have added further text: “From in silico analysis, the peptide TVQIPGGERVPFLF is likely to have anti-diabetic activity, further validating the observed α-amylase inhibitory activity identified in vitro for the hydrolysate. Moreover, several of the peptides are predicted to impart an Umami taste, including RSPTGEIIFGGETM from the Scenedesmus mix hydrolysate. Umami peptides with antioxidant properties may be used as natural preservatives in meat products, sauces, soups and snacks, to extend the shelf-life of foods.”
Comment 3: In conclusions: the sentence "Such microalgal hydrolysates could provide essential amino acids to consumers as well as tertiary, health benefits to improve human global health." should be changed or deleted. The authors did not provide amino acids profiles so this is an unverified hypothesis.
Response 3: We agree with the reviewer and the sentence now reads as follows:” Hydrolysates generated demonstrate anti-ACE-1, anti-amylase and antioxidant activities. They also have techno-functional properties including emulsifying and OHC and WHC potential. This makes the hydrolysates suitable for use as ingredients to improve product functionality and health benefits. The identified bioactive peptides have predicted anti-inflammatory, anti-diabetic and umami attributes. Such microalgal hydrolysates provides a protein source for consumers that has tertiary, health benefits, which may improve human health. MS combined with in silico analysis and in vitro bioassays are useful tools for the drug discovery stage of novel compound isolation prior to animal and clinical testing.”
Comment 4: Please verify the verb tense in all manuscript.
Response 4: Verb tense in the manuscript is the past tense and language was reviewed using spellcheck.
Comment 5: Please pay attention to the number of table reported in the text (i.e Table 3 or 4 in section about peptides MS analysis).
Response 5: We have checked the number for each table throughout.